# Follow-Up of Men Who Have Undergone Focal Therapy for Prostate Cancer with HIFU—A Real-World Experience

**DOI:** 10.3390/jcm12227089

**Published:** 2023-11-14

**Authors:** Katharina Sophie Mala, Henning Plage, Lukas Mödl, Sebastian Hofbauer, Frank Friedersdorff, Martin Schostak, Kurt Miller, Thorsten Schlomm, Hannes Cash

**Affiliations:** 1Department of Urology, Charité University Medicine Berlin, 10117 Berlin, Germany; henning.plage@charite.de (H.P.); hannes.cash@prouro.de (H.C.); 2Institute of Biometry and Clinical Epidemiology, Charité University Medicine Berlin, 10117 Berlin, Germany; 3Department of Urology, Koenigin Elisabeth Herzberge, 10365 Berlin, Germany; 4Department of Urology, Otto-von-Guericke-University Magdeburg, 39106 Magdeburg, Germany; 5PROURO, 10117 Berlin, Germany

**Keywords:** prostate cancer, focal therapy, HIFU, MRI

## Abstract

Purpose: To determine oncological and functional outcomes and side effects after focal therapy of prostate cancer (PCa) with high-intensity focused ultrasound (HIFU). Methods: This retrospective single-center study included 57 consecutive patients with localised PCa. Aged 18–80 with ≤2 suspicious lesions on mpMRI (PIRADS ≥ 3), PSA of ≤15 ng/mL, and an ISUP GG of ≤2. HIFU was performed between November 2014 and September 2018. All men had an MRI/US fusion-guided targeted biopsy (TB) combined with a TRUS-guided 10-core systematic biopsy (SB) prior to focal therapy. HIFU treatment was performed as focal, partial, or hemiablative, depending on the prior histopathology. Follow-up included Questionnaires (IIEF-5, ICIQ, and IPSS), prostate-specific antigen (PSA) measurement, follow-up mpMRI, and follow-up biopsies. Results: The median age of the cohort was 72 years (IQR 64–76), and the median PSA value before HIFU was 7.3 ng/mL (IQR 5.75–10.39 ng/mL). The median follow-up was 27.5 (IQR 23–41) months. At the time of the follow-up, the median PSA value was 2.5 ng/mL (IQR 0.94–4.96 ng/mL), which shows a significant decrease (*p* < 0.001). In 17 (29.8%) men, mpMRI revealed a suspicious lesion, and 19 (33.3%) men had a positive biopsy result. Only IIEF values significantly decreased from 16 (IQR 10.75–20.25) to 11.5 (IQR 4.5–17) (*p* < 0.001). The rate of post-HIFU complications was low, at 19.3% (11 patients). The limitation of this study is the lack of long-term follow-up. Conclusions: HIFU as a therapy option for nonmetastatic, significant prostate cancer is effective in the short term for carefully selected patients and shows a low risk of adverse events and side effects.

## 1. Introduction

An increase in prostate-specific antigen (PSA) screening has resulted in a larger number of patients being diagnosed with Prostate Cancer (PCa), especially those with more localized and low-risk features [1,2]. For intermediate and high-risk PCa, radical whole-gland therapies such as prostatectomy and radiation therapy are the standard of care, despite the commonly known side effects such as incontinence, erectile dysfunction, and bowel disorders [3,4,5]. For low-risk PCa (PSA < 10 ng/mL, Gleason 3 + 3, cT1-2a), active surveillance (AS) is an established treatment option. Due to the mostly indolent nature of low-risk prostate cancer, cancer-specific mortality after more than 10 years is less than 5% [6,7]. The findings of 15 years of follow-up (FU) data from the ProtecT trial even suggest that early aggressive treatment in low-risk patients can actually result in more harm than good [6]. Some men cross over to active therapeutic options due to cancer progression, but others decide for active treatment for various reasons [6]. Therefore, focal therapy (FT) may present a minimally invasive but precise treatment option if men decline the current standard treatment [8]. FT aims to maintain cancer control while simultaneously decreasing the quantity of side effects obtained by only treating areas of known cancer plus a specified margin. This possibility was enabled by improvements in cancer localization through the introduction of mpMRI and targeted biopsies into the diagnostic pathway [9]. In current cohorts, diagnosed by multiparametric magnetic resonance imaging (mpMRI) followed by targeted biopsy, up to 16.2% of men may be eligible for focal therapy [10]. High-Intensity Focused Ultrasound (HIFU) is the most used technology with the most available data [11,12,13,14,15,16,17], but the studies differ in functional outcomes.

Due to the potentially lower morbidity and impact on urogenital function, the demand for this form of therapy is increasing among patients. Nonetheless, patients should be informed about the current guidelines and recommendations. Before an unrestricted recommendation in favor of incorporating focal therapies into routine clinical practice can be made, we still require robust prospective trials. According to the current EAU Guidelines, AS should remain the primary option for patients with low-risk PCa. Due to a lack of data, HIFU is only recommended for patients with intermediate-risk PCa within a clinical trial setting or a well-designed prospective cohort study. As a salvage therapy, HIFU should also only be performed on selected patients in experienced centres as part of a clinical trial [18].

The aim of this study is to evaluate the functional and oncological outcomes of HIFU in a single-center investigation.

## 2. Materials and Methods

This retrospective analysis included all men who received focal therapy using HIFU with the Focal One system (EDAP TMS, France) between November 2014 and September 2018 at the Department of Urology Charité—University Medicine Berlin, Germany. All patients were allocated to the medical consultation of the Department of Urology Charité Berlin after the initial diagnosis of prostate cancer. An MRI-targeted biopsy was the standard diagnostic evaluation. All patients were informed about the standard therapies for the treatment of prostate cancer due to the German S3 Guidelines at that time [19]. Patients with Gleason grade 1 were recommended to undergo AS. According to the German guidelines, AS was not recommended if ≥3 cancer cores were positive and the cancer core length was >50%. If patients refuse AS or the standard whole gland therapies, alternative therapies such as HIFU are discussed. Before carrying out focal therapy, a comprehensive briefing was conducted regarding the current data. This included the fact that there is no evidence of the therapy being equivalent to standard treatments. Additionally, patients were informed that any necessary salvage therapy following primary focal therapy may result in poorer functional and oncological outcomes. Before FT, all men signed a written consent form.

The ablation with the Focal One^®^ HIFU Robotic System is a non-invasive procedure using high-performance and high-intensity focused ultrasound (HIFU) technology. The procedure can be performed under general or spinal anesthesia. The patient is positioned on a standard operating table in the lateral decubitus position. The probe is manually inserted into the rectum by the physician before the system acquires a 3D ultrasound volume of the prostate. With the help of elastic fusion of MRI or 3D biopsy maps, the urologist performs precise planning of the target with adequate margins while avoiding critical structures, sparing healthy surrounding tissue, and minimizing side effects. The probe comes with eight focal points, each 5 mm in length, to enable precise treatment of the tumor. The computerized system creates virtual slices of the prostate every 1.7 mm to cover the entire ablation area. For each individual slice, the urologist shapes the outline of the target. The quantity of ultrasound energy rapidly raises the temperature at the focal point, leading to coagulation necrosis without damaging healthy tissue. During the procedure, the urologist maintains control and monitors the ablation through real-time visualization. Pausing and readjusting the plan is possible at any time [20]. After the procedure, we performed a contrast-enhanced ultrasound with SonoVue^®^ to monitor the results of the treatment (Figure 1C). Every FT was executed by the same urologist (H.C.), who is an EDAP TMS-certified HIFU trainer.

Where available, PCa was diagnosed by an MRI/US fusion-guided targeted biopsy (TB) combined with a TRUS-guided 10-core systematic biopsy (SB). The men received either hemiablation, partial hemiablation (zonal treatment), or treatment of a focal area only, depending on the localization and quantity of the tumor.

The inclusion criteria were: age between 18–80 years, a maximum of two suspicious lesions of the prostate according to the Prostate Imaging Reporting and Data System (PI-RADS) in mpMRI classified as ≥3 (Figure 1A), local low or intermediate-grade PCa, PSA of ≤15 ng/mL, and an ISUP GG of ≤2. Individual aberrations were accepted after careful consideration.

Before the procedure, patients were asked to complete a questionnaire, including the International Consultation of Incontinence (ICIQ) Score, the International Prostate Symptom Score (IPSS), and the International Index of Erectile Function (IIEF-5) Score, prior to focal treatment. A post-HIFU mpMRI at 6–12 months (Figure 1B) and confirmatory biopsy were recommended for all patients.

Follow-up data were collected between December 2019 and February 2020 and included clinical parameters, follow-up mpMRI, subsequent prostate biopsy, ICIQ, IPSS, and IIEF-5-Score, and the complication rate within the first 30 days and thereafter, according to Clavien–Dindo. Collecting patient data and conducting follow-ups received approval from the local institutional review board of Charité Berlin (EA1/004/22). All activities were carried out in accordance with the Helsinki Declaration.

Wilcoxon’s Rank Sum Test was used for the analysis of significant differences in patient-reported outcome measures, such as ICIQ, IIEF-5, and IPSS, PSA level, Gleason Score, and PI-RADS Score, before and after treatment.

The data were analyzed using the Statistical Package for the Social Sciences (SPSS^®^) Software Version 29.0 for Windows (SPSS Inc., IBM Company, Chicago, IL, USA).

Statistical significance was considered to be *p* < 0.05.

## 3. Results

Between November 2014 and September 2018, 57 patients were treated with HIFU, of which 26 (44.8%) men received a Hemiablation, 14 (24.1%) men received a partial Hemiablation, 2 (3.4%) men received bifocal therapy, and 15 (25.9%) men received focal therapy. The median age of the cohort was 72 years (IQR 64–76). The latest median PSA value before HIFU treatment was 7.3 ng/mL (IQR 5.75–10.39 ng/mL). The median prostate volume was 47 mL (IQR 33.5–60 mL). The patient baseline features are displayed in Table 1. 56 patients (98.2%) received an mpMRI prior to the procedure. One man had to abstain from it due to an implanted pacemaker. The PI-RADS distribution was as follows: PI-RADS 2 2 (3.6%) patients, PI-RADS 3 6 (10.7%) patients, PI-RADS 4 34 (60.7%) patients, and PI-RADS 5 11 (19.6%) patients. The ISUP GG of all SB/TBs was 1 in 33 (57.8%) patients, 2 in 21 (36.8%) patients, 3 in 1 (1.8%) patient, and 4 in 1 (1.8%); please see Table 2.

The median follow-up was 27.5 (IQR 23–41) months, and 50 men (87.7%) completed the questionnaires. The HIFU led to no significant change in continence (ICIQ, *p* = 0.072) or micturition (IPSS, *p* = 0.991). The IPSS life quality index significantly improves (*p* = 0.049). A significant decrease was seen in IIEF values, from 16 (IQR 10.75–20.25) to 11.5 (IQR 4.5–17) in the ability to gain and maintain an erection (*p* < 0.001). The results of the functional FU are shown in Figure 2.

The median PSA value at the time of the follow-up was 2.5 ng/mL (IQR 0.94–4.96 ng/mL), which shows a significant decrease (*p* < 0.001) in PSA value compared to the PSA value before HIFU treatment of 7.3 ng/mL (IQR 5.75–10.39 ng/mL). A post-HIFU mpMRI was recommended to all patients and was performed in 36 (61.4%) men after a median of 12.5 (IQR 8.25–23.5) months post-focal therapy. In 17 (29.8%) men, mpMRI revealed a suspicious lesion. Table 2 shows the PI-RADS distribution. A total of 10 (17.5%) lesions were detected in-field (PI-RADS 3 3 (5.3%) patients, PI-RADS 4 5 (8.8%) patients, and PI-RADS 5 2 (3.5%) patients) and 7 (12.3%) out-field (PI-RADS 4 6 (10.5%) patients and PI-RADS 5 1 (1.8%) patients). In 18 (31.6%) men, no suspicious lesion was found (PI-RADS ≤ 2). A total of 26 (45.6%) patients received a biopsy, resulting in 19 (33.3%) men with a positive biopsy result: ISUP GG 1 12 (21.1%) patients, ISUP GG 2 3 (5.3%) patients, ISUP GG 4 3 (5.3%) patients, and ISUP GG 5 one (1.8%) patients. A total of 8 (15.8%) positive biopsy results were located in-field, 4 (7%) out-field, and 6 (10.5%) were located in-field as well as out-field. The detailed results of cancer recurrence after HIFU are shown in Appendix A. A total of 7 (12.3%) patients had a negative biopsy result. The 31 (54.4%) patients not receiving a biopsy had either only a PI-RADS-2 lesion in the post-HIFU mpMRI, a low PSA, or simply refused to take part in the common follow-up. The patient who did not undergo a pre-treatment MRI did not undergo a follow-up MRI, i.e., due to a pacemaker, but received PSA testing and a high-resolution ultrasound biopsy with the ExactVu™ micro-ultrasound system.

Approximately 15 (26.3%) men underwent salvage treatments, with 2 men receiving more than one therapy. A total of 3 men are currently under active surveillance; 5 men underwent radical prostatectomy; 4 men received radiation therapy; 4 men received androgen deprivation therapy (ADT); and one man received re-focal therapy with irreversible electroporation (IRE). The median time to salvage treatment was 17 months (IQR 12.5–24.5 months).

The median catheter dwell time was 2 days (IQR 1–3 days), and the median hospital stay was 4 days (IQR 3–4 days). In 10 (17.5%) patients, the dissection was made with an indwelling catheter.

The rate of post-HIFU complications was low, at 19.3% (11 patients). A total of 10 patients experienced adverse events (AE) within the first 30 days, all classified as ≤2 according to the Clavien–Dindo classification: 4 (7%) suffered from postoperative urinary retention (UR) and had to be recatheterized; 4 (7%) had a urinary tract infection (UTI) and needed to be treated with antibiotics; and one (1.8%) patient reported anal pain during defecation. One (1.8%) patient had hematuria for about 6 months and was included in both complication groups. One (1.8%) patient had a Clavien–Dindo-classed 1 AE more than 30 days after the procedure; please see Table 3.

## 4. Discussion

The main aim of minimally invasive focal therapy is to minimize side effects by treating only a limited area of tissue without affecting the whole gland or resorting to surgery. Next to oncological outcomes, the patient reported outcomes PROMS are crucial in healthcare and clinical research. It helps to measure the quality of life and the effectiveness of medical treatment from the patient’s point of view. One of the strengths of this study is the high rate of patient-reported outcomes (PROM) returns (87%). We observed no significant changes in micturition and continence, as shown in the results of the IPSS and the ICIQ-score. In contrast, after radical treatments, the rates of incontinence with more than one pad per day in the prostatectomy group were up to 18% at 7 years and 24% after 12 years [21]. In our cohort, only the ability to gain an erection was affected by HIFU. The IIEF value showed a median decrease from 16 down to 11.5, which can still be seen as a significant improvement compared to radical prostatectomy, which reports impotence and erectile dysfunction in up to 80% of the patients [21]. Comparing our results to other studies that examined HIFU as a focal therapy option is challenging due to the utilization of varying scores and questionnaires. For measuring continence, the Expanded Prostate Cancer Index Composite (EPIC) urinary domain questionnaire or the grade of stress urinary incontinence (SUI) questionnaire were used [12,16,17]. No other group used the ICIQ for HIFU. Nonetheless, Guillaumier et al., showed similar results, with 100% of patients achieving social continence and no men requiring more than 1 pad per day [17]. Ganzer et al., showed slightly more SUI, with 2.4% of patients with grade 2 SUI needing more than 1 pad per day and 0.7% of patients with grade 3 SUI requiring intervention; 83.1% of patients were pad-free [12]. Three studies also used the IIEF-5 questionnaire to detect erectile dysfunction. Hardenberg et al., showed a decrease of 2 points, and Rischmann et al., showed a decrease of 1.2 points, and in the HEAT trial, only 0.4 points were detected 12–24 months after HIFU [13,22,23]. In contrast to other studies, our results showed a greater decrease, with four points less prior to the intervention. However, compared to the other studies, the follow-up was conducted later and our cohort was older, with a median age of 72 years (IQR 64–76), compared to a median age of 70 years (IQR 52–78) [22], 66.4 years [23], and 64.9 years (IQR 61–69) [13].

Although post-HIFU biopsy was recommended to all men, only 26 (45.6%) patients underwent this procedure. PCa was detected within the treated area, outside the treated area, and in both areas. Of the 14 positive biopsies within the treated area, 4 had a Gleason Score of 6 with a maximum cancer core length of ≤3 mm and showed a decrease in size/Gleason compared to the pre-HIFU biopsy, which can be considered clinically acceptable with no further need for therapy [24]. Other studies show similar results of positive biopsies: Rischmann et al., 33%, Ganzer et al., 25.6%, and Nahar et al., 30% [12,13,16]. We need to question the comparability as all studies had a different collective of patients, treatment, and follow-up. All three, unlike our cohort, partly combined HIFU with transurethral resection of the prostate (TURP) and had a higher quantity of post-HIFU biopsies, with 91% for Rischmann et al., 55.2% for Ganzer et al., and 58% for Nahar et al. [12,13,16]. Furthermore, Rischmann et al., and Ganzer et al., included a higher percentage of GG-1 cases in their studies [12,13]. The rate of approx. 40% of ISUP ≥ 2 patients in our cohort might also explain the high rate of relapse in 26%, as these cancers could be more aggressive. The rate of salvage treatment differs in the literature from 10–40% depending on the FU period [12,13,14,22,25]. The HEAT multicenter study in the UK with similarly higher percentages of intermediate and high-risk cohorts shows a high failure-free survival (FFS) of 90% after 3 years FU [14]. Nonetheless, over 7 years, approximately 25% of the cases needed a second session of focal HIFU, which was not labeled as “salvage therapy.” In contrast to our cohort, Duwe et al., even recognized lower FFS, with 62% at a median FU of 23 months. The Heterogeneity of this study population with varying risk profiles, different definitions of “failure-free survival,” and FU periods is often one of the main reasons that could lead to these variations in outcomes. Advancing the field of FT requires the establishment of standardized nomenclature and the evaluation of treatment outcomes [26]. It might also be possible that some of our patients with ISUP 1–2 could have been underdiagnosed in the initial diagnosis regarding the high rate of PIRADS 5 lesions (ca. 20%). As we know, accurate risk stratification is essential for making the appropriate treatment decision. Therefore, beyond the information about MRI, PSA, and ISUP, additional predictive tools that offer a better understanding and alignment of the tumor’s phenotype with its genotype are required [14]. Other rationales for the heterogeneous outcomes of failure-free survival could be the experience of the physician as well as different HIFU technologies, but in most studies, the FocalOne system was used.

When looking at post-HIFU complications classified according to the Clavien–Dindo system, studies by Guillaumier et~al. (2018) displayed similar results; Schmid et~al. (2019); and Nahar et~al. (2020) experienced slightly more complications, as shown in Table 3 [15,16,17,25]. Overall, our patients only experienced ≤2 complications in the Clavien–Dindo classification, and HIFU can be seen as a safe treatment option. Overall, we detected only complications (Clavien–Dindo Grade ≤ 2), indicating that HIFU can be considered a safe treatment option.

A major limitation of this study is its retrospective nature and the fact that it only presents the experience of a single center. According to the current EAU guidelines, focal treatment with HIFU is recommended in the context of a prospective registry that future treatments should consider. Currently, in Germany, there is an internet-based database for recording the required parameters, the so-called “@-registry.” Our data were added to this database. However, surveys within the urological communities in Western countries reveal that approximately half of European urologists and one in four urologists in the United States recommend and carry out FT, even beyond clinical trials [26]. With a median FU of 27 months, we do not achieve long-term FU in contrast to other HIFU trials [14]. So far, the updated results of the multicentre registry study HEAT in the UK have shown the most long-term effects of HIFU in a large cohort. Although the follow-up time of ≥5 years was 82 months, the median overall follow-up time of only 32 months is comparable to our retrospective study. As HIFU is a relatively new procedure, we unfortunately do not have data from robust prospective trials such as PIVOT or ProtecT [6,27]. At the moment, there are ongoing randomized controlled trials, such as IP4-CHRONOS and PART, that are comparing radical treatment approaches to focal therapy. These studies aim to strike a balance between clinicians and patients. However, if they succeed in recruiting participants, it will still take approximately another decade before the primary outcomes are determined [28,29]. Another limitation that must be mentioned is compliance with medical advice. Although all men were recommended to undergo a post-HIFU mpMRI at 6–12 months followed by a confirmatory biopsy, independent of the mpMRI result, unfortunately, compliance with the recommendation was low.

Despite these limitations, the results of our analyses contribute to a better understanding of the use of HIFU in real-world situations.

## 5. Conclusions

This study showed that HIFU is a safe and minimally invasive focal therapy option for carefully selected patients with clinically significant localized PCa at low to intermediate risk. We saw a low probability of urinary side effects and moderate erectile reduction compared to radical whole gland therapy.

## Figures and Tables

**Figure 1 jcm-12-07089-f001:**
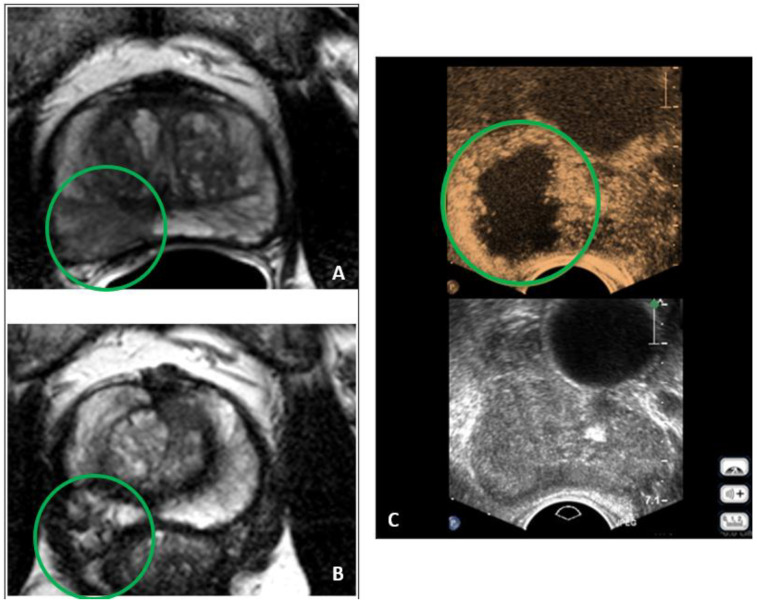
It shows a 60-year-old patient in the cohort. (**A**) mpMRI showing PCa before treatment with HIFU; (**B**) mpMRI showing the former PCa area 6 months after the treatment; (**C**) post-treatment contrast-enhanced ultrasound.

**Figure 2 jcm-12-07089-f002:**
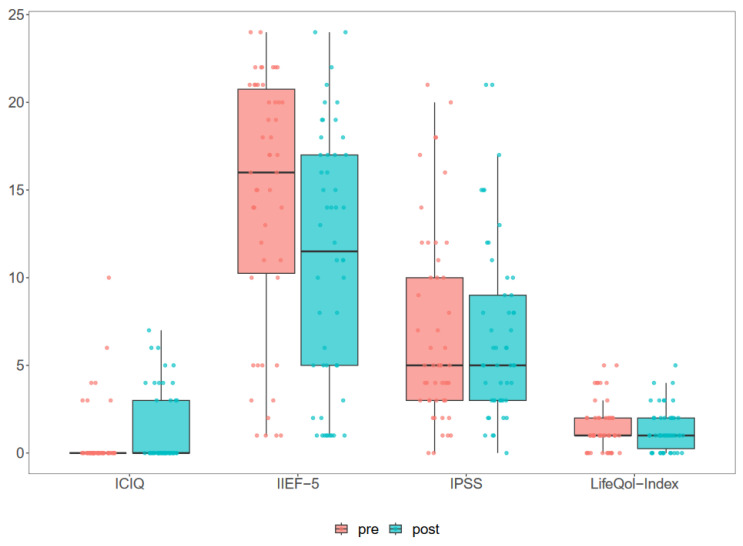
Patient related outcome measures (PROM) relating continence, erectile function, micturition and quality of life.

**Table 1 jcm-12-07089-t001:** Patients characteristics pre HIFU and at time of Follow Up.

	Pre HIFU		Time of Follow-Up	
	Median	IQR	Median	IQR
N = 57			N = 52	
Age (years)	72	64–76	76	66.25–78
FU interval/months			27.5	23–41
Prostatavol/mL	47	33.5–60		
			N = 53	
PSA ng/mL	7.3	5.75–10.39	2.5	0.94–4.975

**Table 2 jcm-12-07089-t002:** Association between biopsy and imaging results and the HIFU application method.

	Value	%							
Bifocal	2	3.4							
Focal	15	25.9							
Hemi	26	44.8							
Part Hemi	14	24.1							
**Pre HIFU**		**Partial hemi (n = 14)**	**Hemi (n = 26)**	**Focal, Bifocal (n = 17)**	**Post HIFU**		**Partial hemi**	**Hemi**	**Focal**
					Biopsy	N = 26	6	10	10
Biopsy (ISUP)	1	10 (71.4%)	13 (50%)	10 (58.8%)	ISUP	1	5	4	3
	2	4 (28.6%)	10 (38.5%)	7 (41.2%)		2	0	0	3
	3	0	1 (3.8%)	0		3	0	0	0
	4	0	1 (3.8%)	0		4	1	2	0
	unknown	0	1 (3.8%)	0		5	0	1	0
						in-field	2	4	2
						Out-field	3	0	1
						In and out-field	1	3	2
						none	0	3	5
					mpMRI	N = 36	11	14	11
mpMRI (PI-RADS)	2	0	2 (7.7%)	1 (5.9%)	treated area	≤2	8	9	8
	3	1 (7.2)	5 (19.2%)	0		3	0	2	1
	4	10 (71.4%)	14 (53.8%)	10 (58.8%)		4	1	2	2
	5	3 (21.4%)	3 (11.5%)	5 (29.4%)		5	2	0	0
	unknown		2 (7.7%)	1 (5.9%)	Untreated area	4	1	1	4
						5	1	0	0

**Table 3 jcm-12-07089-t003:** Postoperative complications after HIFU in comparison to other study cohorts.

	Charité Berlin	F.A. Schmid et al. [15]	Guillaumier et al. [17]	Nahar et al. [16]
**Patients (n)**	57	98	625	52
**Complications ≤ 30 d**	10 (17.5%)	35 (35.7%)		
**Complications > 30 d**	1 (1.8%)	2 (2%)		
**Total complications**	11 (19.3%)	37 (37.8%)	127 (20.3%)	32 (62.3%)
**Complications**	**Value**	**%**		
**Total**	11	19.3		
**Within <30 days**	10	17.5		
**Urinary retention**	4	7		
**Urinary tract infection**	4	7		
**Macrohematuria**	1	1.8		
**Anal pain**	1	1.8		
**After 30 days**	1	1.8		
**Macrohematuria**	1	1.8		

## Data Availability

All data generated or analyzed during this study are included in this published article and its Appendix A.

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
