# Peer review of "Follow-Up of Men Who Have Undergone Focal Therapy for Prostate Cancer with HIFU—A Real-World Experience"

_jcm, 2023, doi:10.3390/jcm12227089_

Round 1
Reviewer 1 Report
Comments and Suggestions for Authors
This study demonstrates good understanding of functional outcomes following partial gland ablation for localised prostate cancer.
Given the recent publication of outcomes from the Protect trial, the short follow up time is of particular concern and is not addressed in the discussion. Further the limited follow up period is a concern given the most recent patients were treated 5 years ago. According to EAU guidelines, focal treatment with HIFU is permitted only within the context of a prospective registry, thus longer follow up would be anticipated. Further the 26% retreatment rate appears high given the median follow up.
Finally, the majority of patients had grade group 1 disease, as such would be expected to be offered active surveillance over any other treatment- suggesting either these patients were underdiagnosed, or all patients with intermediate risk disease required further treatment.
I would suggest reconsideration following analysis with updated follow up, and secondary analysis of the intermediate risk group patients.
Reviewer 2 Report
Comments and Suggestions for Authors
Congratulations on the present study! HIFU therapy represents one of the promising ways to manage prostate cancer.
There are some aspects that I consider it can be improved.
1. Please insert some guideline recommendations in the introduction.
2. Please explain how the authors selected the patients for HIFU.
3. Did the patients expressed their informed consent. Was the declaration of Helsinki respected?
4. How did the authors evaluated the relapse?
5. All the patients were treated by the same therapist?
Comments on the Quality of English Language
The English language is fine!
Round 2
Reviewer 1 Report
Comments and Suggestions for Authors
Thank you for addressing the primary concerns of short follow up duration and recurrence rates. A strength of the study that should be commended is the high rates of PROM returns.
On review of the amendments further questions arise.
- Was the eligibility for focal therapy truly any patient 18-80 years old? It is remarkable to consider any patient <50 years old to be suitable given the lack of long term efficacy data globally.
- The majority of patients treated had ISUP group 1 disease- the authors state in the introduction that these patients should be offered AS- what was it about these cases that led to them having treatment?
- Can the authors please confirm what the biopsy regime and follow up was for the patient that did not undergo pre-treatment MRI?
- Can the authors please elaborate on what is meant by bi-focal and partial hemiablation? This would be better done pictorially.
- Is the use of figure 1d referenced anywhere in the manuscript? If not please remove.
- Table 2 reports 36 patients underwent post-treatment MRI however the manuscript reports 35. Further the manuscript reports 9 patients had out-of-field disease on post-treatment biopsy, table 2 reports 8, please clarify. I think the authors have typo’d the table reference to report in table 5 (doesn’t exist).
- How did the authors determine which patients had post-treatment biopsies (ie the discussion discussed all patients were recommended biopsy but there was a low uptake- <50%) and how were those biopsies performed? MRI target only vs target and systematic vs systematic only?
- Table 2- can the authors please confirm if any patients had areas of concern on MRI in the treated AND untreated area? Currently it reads as though this is not the case.
- Can the authors elaborate on the pre-treatment histology of the areas later found to have in field and out-of field recurrence? Was there any upgrading to suggest this wasn’t captured pre-treatment, or ongoing similar disease that wasn’t completely treated despite use of contrast enhanced US at the time of treatment?
- The median hospital stay was 4 days, can the authors please explain whether this is anticipated? Within the USA and UK patients are typically discharged the same or following day.
- Figure 2 is a simplistic representation of the authors’ primary outcome. It doesn’t clearly represent the outcomes stated in the manuscript (IPSS QoL improvement shows as 1 pre and post treatment). I would recommend box and whisker plots reporting or individual plots of median scores of each domain reported. It is important to know the timeframe of post-treatment questionnaires as the Index trial does report improvement of function up to a year following treatment.
- Can the authors elaborate on how many patients went on to have management of their erectile function post-treatment (PDE5 inhibitors etc)?
- Can the authors provide a time to salvage treatment? Was this within 6 months or towards the end of their follow-up period?
- Can the authors provide a FFS estimate- the discussion reports similar outcomes to previous EDAP studies, however ‘relapse rates’ cannot be compared to FFS.
- Typos noted: ‘sta-tistical’, ‘dayn’
